# Increasing Uptake of Maternal Pertussis Vaccinations through Funded Administration in Community Pharmacies

**DOI:** 10.3390/vaccines10020150

**Published:** 2022-01-20

**Authors:** Anna S. Howe, Natalie J. Gauld, Alana Y. Cavadino, Helen Petousis-Harris, Felicity Dumble, Owen Sinclair, Cameron C. Grant

**Affiliations:** 1Department of General Practice and Primary Health Care, University of Auckland, Auckland 1023, New Zealand; h.petousis-harris@auckland.ac.nz; 2School of Health Sciences, University of Canterbury, Christchurch 1023, New Zealand; 3Department of Paediatrics, Child and Youth Health, University of Auckland, Auckland 1023, New Zealand; n.gauld@auckland.ac.nz (N.J.G.); cc.grant@auckland.ac.nz (C.C.G.); 4School of Pharmacy, University of Auckland, Auckland 1023, New Zealand; 5Department of Epidemiology and Biostatistics, School of Population Health, University of Auckland, Auckland 1023, New Zealand; a.cavadino@auckland.ac.nz; 6Waikato District Health Board, Hamilton 3240, New Zealand; Felicity.Dumble@waikatodhb.health.nz; 7Child, Women and Family Services, Waitakere Hospital, Waitemata District Health Board, Auckland 0610, New Zealand; Owen.Sinclair@waitematadhb.govt.nz; 8General Paediatrics, Starship Children’s Hospital, Auckland District Health Board, Auckland 1023, New Zealand

**Keywords:** pharmacists, immunisation, pregnancy, vaccination, vaccination coverage

## Abstract

Although maternal pertussis vaccination is recommended, uptake is suboptimal in New Zealand (NZ), despite full funding in general practice and hospitals. We determined whether funding maternal pertussis vaccination in community pharmacy increases its uptake. Pertussis vaccination during pregnancy was compared between non-contiguous, demographically similar regions of NZ. The pertussis vaccine was funded at pharmacies from Nov 2016 in one NZ region (Waikato), but not in comparator regions (Northland, Hawkes Bay). Vaccinations during pregnancy were determined from the National Immunisation Register, general practice and pharmacy claims data, and a maternity database. Comparisons were made using adjusted odds ratios (OR) and 95% confidence intervals (CI) for Nov 2015 to Oct 2016 versus Nov 2016 to Oct 2019. The odds of pregnancy pertussis vaccination increased in the post-intervention versus pre-intervention period with this increase being larger (*p* = 0.0014) in the intervention (35% versus 21%, OR = 2.07, 95% CI 1.89–2.27) versus the control regions (38% versus 26%, OR = 1.67, 95% CI 1.52–1.84). Coverage was lower for Māori versus non-Māori, but increased more for Māori in the intervention versus control regions (117% versus 38% increase). It was found that funding maternal pertussis vaccination in pharmacy increases uptake, particularly for Māori women. Measures to increase coverage should include reducing barriers to vaccines being offered by non-traditional providers, including pharmacies.

## 1. Introduction

Pertussis (whooping cough) causes hospitalisations and deaths, particularly of young infants [1,2], and disproportionately affects indigenous infants [1,3]. Tetanus-diphtheria-acellular pertussis (Tdap) vaccination during pregnancy provides protection to this vulnerable young infant age group, preventing hospitalisations [4] and deaths [5].

Although maternal pertussis vaccinations are recommended and funded in many developed countries [6,7], uptake is often suboptimal [8,9,10], particularly in higher deprivation groups [10,11], including in New Zealand (NZ) [10]. Lack of awareness, misperceptions and safety concerns hinder the uptake of maternal vaccinations [12,13,14]. While healthcare professional recommendations and/or the endorsement of the safety of maternal vaccinations encourages uptake [9,12,14], some antenatal carers do not provide this [9,14,15].

A convenient location for the vaccination event aids uptake during pregnancy [16], for example, provision by midwives or other antenatal care providers at antenatal appointments (often with associated education) [17,18,19]. However, the location of delivery of maternal vaccinations varies within and between countries, and antenatal providers might not vaccinate [20]. In NZ, for example, many antenatal providers work independently, and for logistical reasons (e.g., cold chain) are unlikely to provide maternal vaccinations [21].

Administering vaccines in pharmacy is associated with increased uptake [22], and pharmacists have been proposed to provide funded maternal vaccinations [14,23]. United States (US) hospital pharmacists administering Tdap vaccines to close contacts of neonates increased Tdap vaccination substantially [24]. Some Canadian provinces have maternal influenza vaccination, and, more recently, maternal pertussis vaccination has been funded in pharmacy [25], but this appears uncommon internationally. Whether funded administration of maternal vaccinations in pharmacy improves coverage is unknown.

Our study aimed to determine whether funding and promoting pertussis vaccines in pregnancy in pharmacies over three years would: (1) increase vaccine uptake in pregnant women; and (2), increase uptake in pregnant women of indigenous ethnicity or those living in more socioeconomically deprived areas. For this study of pertussis vaccination, Waikato DHB was the intervention region and Northland and Hawke’s Bay DHBs were control regions, chosen for being non-contiguous with Waikato DHB but with similar demographics. The intervention region includes approximately 420,000 people [26], and the control regions combined approximately 345,000 people [27].

## 2. Materials and Methods

### 2.1. Study Design

This study was a before and after interventional analysis which measured the impact of extending funding for maternal pertussis and influenza vaccinations to community pharmacies through quantitative methods and qualitative interviews [28]. Here, we present quantitative pertussis vaccination results. The New Zealand Ministry of Health Northern B Health and Disability Ethics Committee granted approval (18/NTB/43). The Australian New Zealand Clinical Trials Registry registered the study (ACTRN12616001453471p).

### 2.2. Study Setting

NZ has universal maternity care and selected vaccinations without patient charge, and general practice visits and pharmaceuticals with variable patient co-payments. Pregnant women then usually choose a Lead Maternity Carer (LMC) for the pregnancy, birth care, and infant care up to age six weeks. The LMC is usually an independent midwife or (less frequently) a public hospital midwife. The LMC is infrequently a general practitioner or obstetrician, and a small proportion of women do not access care during their pregnancy.

Vaccinations have become increasingly available in NZ community pharmacies from 2010 [29], starting with influenza vaccination and followed by Tdap vaccine in 2013 [30], while government funding for maternal pertussis vaccinations through general practice and hospitals started in 2013. Maternal pertussis vaccination became funded through pharmacies (between 28–38 weeks’ gestation) in one region (Waikato DHB) in November 2016, as part of this study.

### 2.3. Study Population and Sample

The study population consisted of all pregnant women in the maternity collection residing in the Waikato, Northland and Hawke’s Bay DHB regions, with a surviving fetus beyond 20 weeks’ gestation during the pre- or post-intervention period. As of 2011, coverage for the maternity collection was estimated to be around 80% of known births, as DHB-employed midwifery teams do not have to submit claims to the Ministry of Health [31]. The pre-intervention period was from 1 November 2015 to 31 October 2016 and the post-intervention period from 1 November 2016 to 31 October 2019. Pertussis vaccination during pregnancy in NZ was, for most of the duration of the study, only funded during 28 to 38 weeks’ gestation (extended to the second or third trimester from 1 July 2019 [32]). As such, a woman was considered pre-intervention if the entirety of her 28–38 weeks’ gestational period was within the pre-intervention period, and considered post-intervention if the entirety of her 28–38 weeks’ gestational period was within the post-intervention period.

In order to limit potentially erroneous data, the following exclusion criteria were applied. A woman was excluded if she was younger than 12 years or older than 50 years. In addition, as the maternity collection does not routinely capture loss of a fetus before 20 weeks’ gestation, women were excluded if the pregnancy gestation was less than 20 weeks. Women were also excluded if gestation was greater than 43 weeks, as in NZ a woman would typically have been induced by at least 43 weeks’ gestation.

For the primary analysis, only a woman’s first pregnancy during the study period was included. A secondary analysis was undertaken including all eligible women’s pregnancies during the study period.

Pharmacists in all three regions could attend a training evening on maternal pertussis and influenza vaccinations in October 2016 (Waikato) or in 2017 (Hawkes Bay and Northland). Promotion to pharmacies and consumers for six months from April 2018 (after research funding contracts were finalised) included Facebook posts, posters, and information for pharmacy staff and midwives (Appendix A).

### 2.4. Data Sources

Existing data sources were linked using the National Health Index (NHI) number, a unique identifier for each person in all databases, encrypted by the Ministry of Health to maintain confidentiality (see Appendix A for more information about data sources). The data sources were:National Maternity Collection for women using publicly funded maternity/newborn services.National Health Index (NHI) number for demographic information.National Immunisation Register (NIR) for National Immunisation Schedule vaccinations administered.Proclaim for general practice and pharmacy claims for vaccination events.DHB Maternity Clinic Data for vaccination events within the hospital.Pharmaceutical Collection for community pharmacy dispensing of prescribed medicines.Waikato Pharmacy Claims Data for Tdap vaccination of pregnant women.

### 2.5. Outcomes

The primary outcome was a binary variable: the receipt of Tdap vaccination during pregnancy. Vaccination receipt status for each woman was determined as having a valid entry for a pertussis vaccine in any one of the four data sources during their eligible pregnancy. The data sources where pertussis vaccination could be recorded were prioritised: NIR, Proclaims, Waikato Pharmacy Claims and lastly DHB maternity clinic data. A vaccination was valid if it occurred between the last menstrual period and censoring (delivery or end of assigned study period—whichever came first). A woman’s valid vaccination was assigned to pre- or post-intervention.

### 2.6. Co-Variates

Ethnicity was prioritised (Māori, Pacific, Asian, Other, and NZ European) and categorized as Māori or Non-Māori. Māori are NZ’s indigenous people, comprising 16% of the NZ population at the 2018 census [33]. Socio-economic deprivation was measured using an area level measure (NZ Dep 2013 Index of Deprivation), which was categorized from deciles (‘1’ = least deprived and ‘10’ = most deprived) into quintile (1–5) and binary (low = deciles 1–5 and high = deciles 6–10) variables. The NZ Dep 2013 Index of Deprivation is a small area measure of deprivation that is derived from variables collected at the national census which describe household socioeconomic characteristics [34].

### 2.7. Statistical Analysis

Demographic variables (deprivation, ethnicity, and DHB) missing from the maternity dataset were obtained using the NHI dataset where possible, and missing gestational age or last menstrual period date were imputed using delivery date. Descriptive variables were presented as counts and percentages. Differences between descriptive variables were examined using Chi-square tests, or Fishers exact tests, for differences between control and intervention regions in each study period (pre- and post-intervention). Vaccination status was additionally described as a percentage change (([post-intervention − pre-intervention]/pre-intervention) × 100), for the three study regions and also for all other DHBs, in order to put the results into the context of all of NZ. Association between region (intervention versus control) and Tdap vaccine receipt was described using logistic regression for the primary analysis. A secondary analysis including all of a woman’s eligible pregnancies during the study period was undertaken using generalised estimating equations with a binomial distribution and logit link, and an autoregressive correlation structure accounting for repeated pregnancies. All models were adjusted for maternal age at last menstrual period, maternal ethnicity (Māori/Non-Māori), and maternal deprivation (low/high). An interaction term of region (intervention/control) by time (pre-/post-intervention) was included to test if the Tdap intervention was effective. To examine ethnicity and deprivation differences for Tdap, a three-way interaction term was additionally included, for region (control/intervention) by time (pre-/post-intervention), and either ethnicity (Māori/Non-Māori) or deprivation (low/high). All tests for assessing statistical significance were two-sided with a *p* < 0.05. All statistical analyses were undertaken using SAS Enterprise Guide (9.4) statistical software (SAS Institute Inc., Cary, NC, USA).

## 3. Results

### 3.1. Sample

The final cohort for pertussis vaccination consisted of 27,576 eligible pregnant women in the maternity database in the three study regions for the three-year intervention data, limited by the strict inclusion criteria (Figure 1).

Table 1 describes the demographics for the intervention and control areas. The control regions had proportionally more pregnant women of Māori ethnicity (49% versus 35%, in the pre- and post-intervention periods combined, respectively) and more in the most deprived quintile of deprivation (49% versus 33%, respectively) than the intervention region. Appendix A provides overall demographic information of the three study regions. As of April 2018, telephone contact with all community pharmacies in the three study regions found that 35 of 83 community pharmacies (42%) provided vaccinations in the intervention region, and 28 of 73 pharmacies (38%) in the control regions. Pharmacies in main urban areas were more likely to offer vaccination services than pharmacies outside of those areas (60% versus 21% in the intervention region, respectively, and 48% versus 17% in the control regions, respectively).

### 3.2. Pertussis Vaccine Uptake

In the intervention region, 21% of women received the pertussis vaccine during the pre-intervention period and 35% in the post-intervention period (Table 2). In the control regions, 26% of women received the pertussis vaccine during the pre-intervention period and 38% in the post-intervention period (Table 2). This resulted in a percentage change of 67% in the intervention region (Figure 2) versus 44% in the control regions. This increase in pertussis uptake from the pre-intervention to the post-intervention period was statistically greater in the intervention region than the control region (*p* = 0.0014) (Table 3). As there was no real difference between the crude and adjusted analyses, only the adjusted analyses are presented in Table 3. The secondary analysis, including all of a woman’s eligible pregnancies, did not change the results (data not shown).

Figure 2 shows the increase from pre-intervention to post-intervention in the intervention region and control regions, compared with all other regions in NZ.

Pharmacies delivered 26% (*n* = 1084) of the pertussis vaccinations in the intervention region in the post-intervention period, and 0% (*n* = 7) in the control regions, where women would have needed to choose to pay for administration in pharmacy.

Pertussis vaccine uptake for Māori women was lower than for non-Māori women, in both the pre- and post-intervention study periods (Table 3). The increase in pertussis vaccination uptake for Māori women was greater in the intervention region (OR = 2.42, 95% CI 1.99, 2.97) than the control regions (OR = 1.50, 95% CI 1.29, 1.74) (*p* = 0.0281) (Table 3). There was no significant difference in pertussis vaccine uptake for non-Māori women between the control and intervention regions over the study period (Table 3). There was no significant difference in pertussis vaccine uptake by area-level socioeconomic deprivation, i.e., women living in more socioeconomically deprived areas had a similar proportional increase in uptake to women living in less socioeconomically deprived areas.

A sensitivity analysis undertaken including all pregnancies, not just the first pregnancy of the pre-intervention or post-intervention period, made minimal difference to the findings (data not shown).

The proportion of maternal vaccinations delivered by pharmacy increased over the study period and was particularly evident after the promotional activity which took place from April 2018 to October 2018 (Figure 3). Figure 4 shows that whilst pharmacy-delivered vaccinations increased, this was in addition to, rather than as a replacement for, the vaccinations delivered by other providers.

## 4. Discussion

To our knowledge, this is the first reported study to measure how funding in pharmacy affects the overall uptake of the maternal pertussis vaccination. Funding pertussis vaccination in pharmacy increased uptake modestly overall, and in Māori women (but not in non- Māori women), despite only 42% of pharmacies in the intervention region providing vaccinations. Women in the intervention region had a 67% increase in maternal pertussis vaccination uptake in the post-intervention versus pre-intervention period, while women in the control regions had a 44% increase in maternal pertussis vaccination in the post-intervention versus the pre-intervention period. Pertussis vaccination during pregnancy for Māori women in the control regions increased 38% compared to 117% in the intervention region, between pre- and post-intervention periods. The overall uptake of the maternal pertussis vaccination was higher in the control areas than the intervention area during both study periods (Table 2), but the percentage increase was higher in the intervention area.

The qualitative study which was performed alongside this study, found that most women, general practice staff, midwives and pharmacists supported funded maternal vaccinations in pharmacy because of accessibility, extended hours and there being no appointment requirement [26]. Although pertussis vaccine uptake in Māori women was greater with pharmacy funding, uptake for Māori in pharmacy was relatively low. The pharmacy likely raised awareness for some Māori women (as reported elsewhere [26]), but some may have then attended the medical centre as their usual vaccination place. No significant difference in uptake in non-Māori between intervention and the control regions might reflect a greater awareness of these vaccinations by these women.

The promotional activity in this study was delayed; however, the main result seemed to be a shift from general practice to pharmacy with relatively small gains in overall uptake. This could reflect the fact that many pharmacies providing vaccinations were in urban areas, where there may be less of an access issue than in rural areas, and the fact that the promotion undertaken attracted interest primarily from people well engaged in healthcare already. Promotion to midwives may have increased the likelihood of them suggesting pharmacies rather than general practices for convenience, or to protect pregnant women from exposure to infection while waiting at the doctor’s office, for example [26]. The promotional messages noted both pharmacy and general practice as places for maternal vaccinations.

### 4.1. Comparison with Other Literature

A systematic review investigating community pharmacist administration of vaccinations showed increased vaccine uptake with a pooled analysis relative risk of 2.11 (95% CI 1.63–2.72) [22]. However, maternal vaccinations were not included, and sometimes the benefit did not persist, demonstrating a need for robust research with a parallel control group and sufficient duration, as has been provided in the current study.

In a US hospital study, pertussis vaccination of close contacts of neonates (not pregnant women) increased from 1.3 per month to 85 per month with pharmacist involvement [24], a considerably larger percentage increase than that which occurred in our study. However, the current study examined pregnant women in a community setting with many different providers, who each see relatively few people eligible for maternal vaccinations, so it may be less top-of-mind for the healthcare professionals than in a single centre.

Previous studies using pharmacists administering vaccines as the intervention were identified as having a high risk of bias, due to non-randomised study design, recall bias, selection bias, non-blinding of the outcome assessment, and short follow-up periods [22]. While this study did not use a randomised design, the approach was robust, with the pertussis vaccine intervention applied to one region and compared with geographically distinct but otherwise similar control regions, and objective measurement of vaccines administered across all providers over three years. Individual pharmacies were not invited into the study.

Provision of maternal vaccinations by antenatal care providers can increase uptake [18,19,35,36], sometimes considerably. However, some of the studies reporting this were of short duration [36], and/or involved limited sample sizes [18,36], had a single site or limited numbers of providers for the intervention [18,19,36], and/or had no comparator group [18]. While ideally maternal vaccinations would be available from multiple locations, and particularly from antenatal care providers who could administer them during the antenatal appointment, this might not be practical for all antenatal care providers. For example, in NZ, barriers to midwives providing maternal vaccinations include challenges of administering a vaccine then being called urgently elsewhere, travelling to women’s homes and maintaining cold chain, or purchasing a vaccine fridge and not being funded for this service [21]. Many midwives in NZ work alone or in a partnership, and may visit the woman at home, rather than working in a hospital or larger birthing centre where vaccinations may work better logistically. In this case, pharmacy seems a feasible and appropriate way to improve access.

### 4.2. Implications

Uptake of maternal pertussis vaccination remained sub-optimal, in particular for Māori women. Maternal vaccinations, particularly pertussis vaccination, were relatively new components of maternity care in NZ at the time of the study, so awareness of and trust in these vaccines might still need building for consumers and healthcare providers, and both might need regular reminders regarding these vaccinations. While antenatal care providers providing vaccinations increases uptake [17,18,19,35], the NZ model of care, with mainly independent midwives, would struggle with appropriate storage and transport for vaccines. General practice might not be visited during pregnancy, while in contrast most pregnant women would attend pharmacies for prescriptions, e.g., for folic acid and iodine from their midwives.

Our qualitative research component of this project found multiple barriers to maternal vaccination uptake [21]. The provision of funded maternal vaccinations in pharmacy can help address awareness, provide information, aid opportunistic vaccination, and reduce barriers to access, such as requiring an appointment, Monday to Friday availability only in general practices, and a concern about sitting in a waiting room with sick people. However, many pharmacies have not offered vaccinations; non-vaccinating pharmacy staff may not be sufficiently informed or proactive with maternal vaccinations, and some pharmacies did not offer vaccinations at all opening hours.

Encouraging more pharmacies to provide vaccinations should increase uptake further. Barriers such as set-up costs, general practice negativity concerns, and insufficient qualified staff [27] might particularly affect pharmacies in high deprivation areas without a private market, and rural pharmacies which might have the greatest need given the staffing shortages of nurses in rural general practice [37] and barriers for pregnant women living in poverty [26]. Funding further vaccines in pharmacy, subsidising training for rural pharmacists, and training all pharmacists in vaccination before qualifying could help. Our qualitative research found general practice participants largely supported pharmacies providing funded maternal vaccinations to aid access [26].

Other barriers and enablers need consideration. For example, ensuring early engagement with an LMC to allow time to discuss maternal vaccinations, and overcoming transport issues and other urgent social needs, are likely to help maternal vaccination messaging to be received and sufficient time to get the vaccinations. Underlying beliefs also need to be ascertained and addressed. There is a need to work with midwives, general practice doctors, nurses and receptionists, hospitals, and pharmacy staff—pharmacists and non-pharmacists—to maximise the opportunities to help improve women’s awareness, understanding of and access to maternal vaccinations. The effect of our promotional campaign indicates that a multi-pronged approach, including extending funding of maternal pertussis vaccination to community pharmacies, can help increase uptake.

This study revealed a significant ethnic disparity in maternal pertussis vaccination between Māori and non-Māori women. For the maternal vaccine programme to reduce the Māori pertussis burden, it must address the ethnic disparities found in this report. The uptake of the intervention by Māori women in this study supports addressing systemic barriers to awareness and access rather than assuming a need to change attitudes in order to increase immunisation in Māori women. This is supported by our qualitative interviews of Māori women and health care providers conducted alongside the study (unpublished).

### 4.3. Strengths and Limitations

This study was planned prospectively, included intervention and control regions for Tdap vaccination (with discrete and relatively similar populations and the same promotional activity), a prolonged period of three years post-intervention, and large cohorts (over 20,000). Only including women who were 28–38 weeks’ gestation entirely in the pre-intervention or post-intervention periods avoided overlap in the pertussis sample. Using multiple vaccination datasets aided data capture and avoided self-reporting, and has been used previously [38]. Some vaccination administrations might still have been missed. Some vaccinations at hospitals may have been uncaptured by the NIR or clinic data. The manual process for the Tdap pharmacy claims data resulted in some keying errors and input of claims dates not administration dates, leading to a likely loss of valid data matching the maternal database dates. Therefore, the number of vaccinations delivered in pharmacy to eligible women, and the effect of the intervention for pertussis vaccination is probably understated.

An estimate in 2011 suggested 20% of data was not captured by the Maternal Collection; these women are most likely to be those treated within hospitals alone, as opposed to by independent midwives, general practitioners, or obstetrician/gynaecologists in private practices who are eligible to claim public funds for the pregnancy. It is possible that the women not captured may differ from those who are captured in likeliness to be vaccinated or accessibility to vaccination. For most of the study period, Tdap was funded for weeks 28–38 of gestation; however, from 1 July 2019 to when our capture finished (30 October 2019), the funding changed, allowing second and third trimester funded vaccination (16 weeks-birth). However, we would not expect to see many additional women included in our study because of this programme change, as these changes take time to implement, and the change occurred during the last few months of the study.

## 5. Conclusions

Funding maternal pertussis vaccinations through pharmacy can increase uptake, but uptake currently remains suboptimal. Encouraging more pharmacies to provide vaccinations, particularly those in rural and high needs areas, and encouraging all pharmacy staff to be informed and proactive with maternal vaccination will maximise the impact of this service. Encouraging all health professionals regularly seeing pregnant women to raise awareness and/or offer maternal vaccinations is necessary. Pharmacy provision might be most useful in areas where not all antenatal care providers provide vaccinations and/or where uptake is limited.

## Figures and Tables

**Figure 1 vaccines-10-00150-f001:**
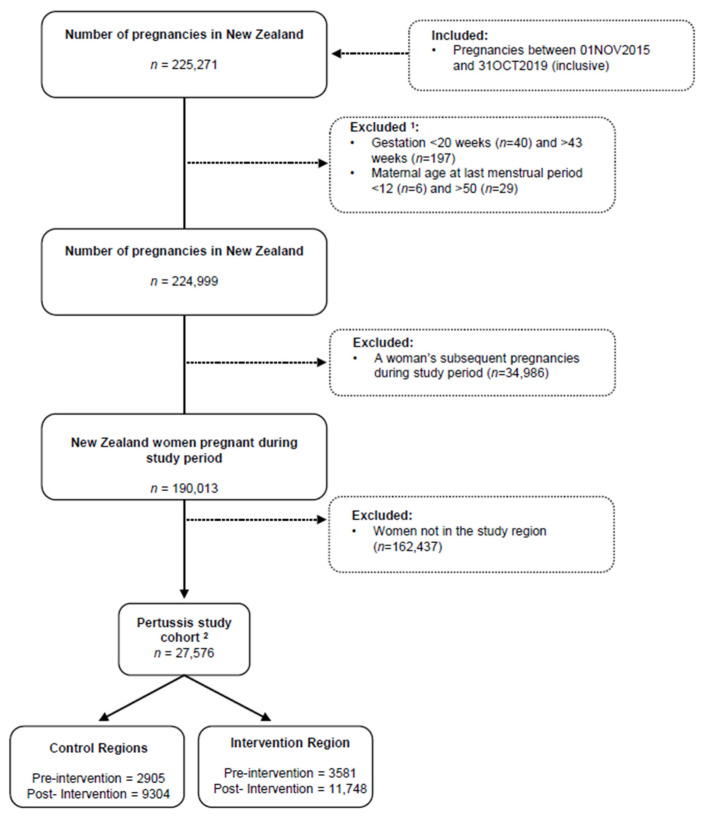
Participant Flow Chart. ^1^ Counts may not necessarily add up as an individual may have multiple exclusion criteria. ^2^ Entirety of 28–38 weeks’ gestational period must be entirely within the pre-intervention period or entirely with the post-intervention period.

**Figure 2 vaccines-10-00150-f002:**
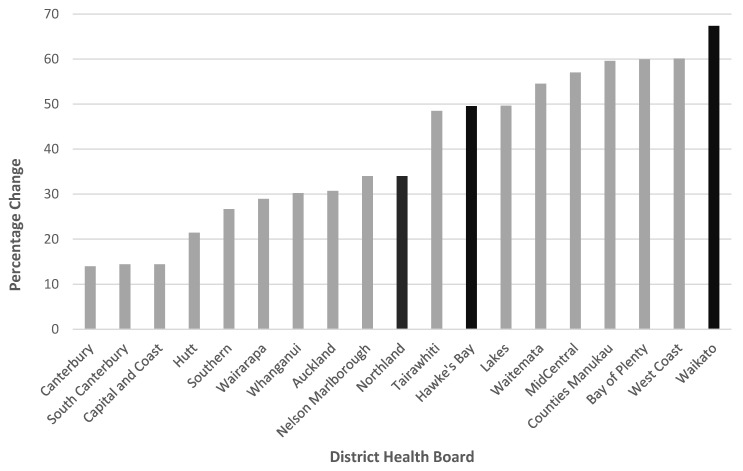
Percentage change in pertussis coverage for pregnant women between pre- and post-intervention periods, by the District Health Board (dark bands are the study regions, Waikato is the intervention region and Hawke’s Bay and Northland the control regions).

**Figure 3 vaccines-10-00150-f003:**
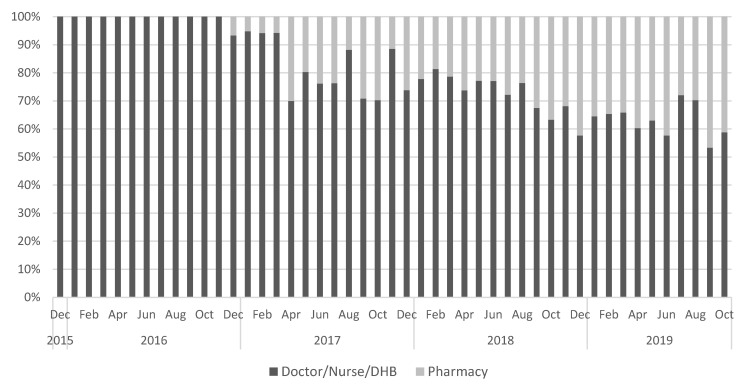
Date of maternal pertussis vaccination in the intervention region, as a proportion administered by pharmacy versus traditional providers.

**Figure 4 vaccines-10-00150-f004:**
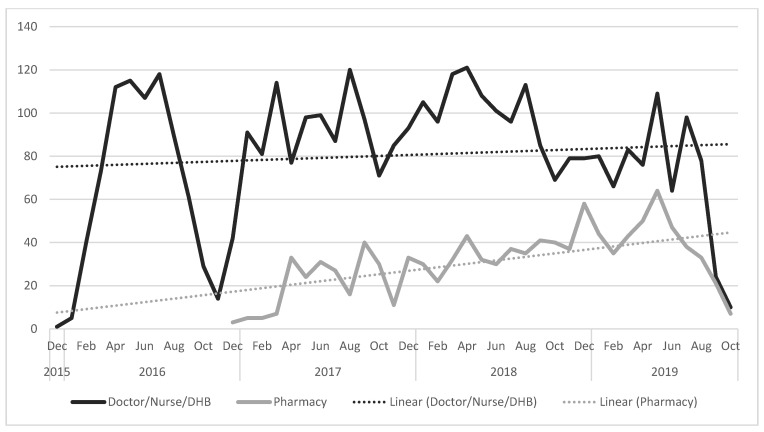
Date of maternal pertussis vaccination in the intervention region, by provider. The drop in vaccination from Sep. 2016 to Feb. 2017 and at the end of the intervention period reflects the requirement to count only women who were weeks 28–38 entirely in the pre-intervention or the post-intervention period.

**Table 1 vaccines-10-00150-t001:** Demographics of women included in the pertussis vaccination cohort who were between 28 and 38 weeks’ gestation between 1 November 2015 and 31 October 2019.

	Total Pertussis Cohort		Pertussis Cohort							
			Pre-Intervention Period				Post-Intervention Period			
			Control Areas		Intervention Area		Control Areas		Intervention Area	
	*n*	(%)	*n*	(%)	*n*	(%)	*n*	(%)	*n*	(%)
Total	27,576	(100.0)	2905	(44.8)	3581	(55.2)	9342	(44.3)	11,748	(55.7)
**Maternal age at LMP (years) ^2,3^**										
12–19	1948	(7.1)	234	(8.1)	239	(6.7)	734	(7.9)	741	(6.3)
20–24	5458	(19.8)	674	(23.2)	760	(21.2)	1885	(20.2)	2139	(18.2)
25–29	8268	(30.0)	845	(29.1)	1079	(30.1)	2733	(29.3)	3611	(30.7)
30–34	7498	(27.2)	704	(24.2)	993	(27.7)	2458	(26.3)	3343	(28.5)
35–39	3508	(12.7)	359	(12.4)	424	(11.8)	1208	(12.9)	1517	(12.9)
40–45	869	(3.2)	86	(3.0)	82	(2.3)	315	(3.4)	386	(3.3)
46+	27	(0.1)	3	(0.1)	4	(0.1)	9	(0.1)	11	(0.1)
**Prioritised Ethnicity ^2,3^**										
Māori	11,302	(41.0)	1534	(52.8)	1289	(36.0)	4429	(47.4)	4050	(34.5)
Pacific	1137	(4.1)	114	(3.9)	146	(4.1)	380	(4.1)	497	(4.2)
Asian	2889	(10.5)	167	(5.7)	430	(12.0)	642	(6.9)	1650	(14.0)
Other	457	(1.7)	31	(1.1)	59	(1.6)	84	(0.9)	283	(2.4)
European	11,791	(42.8)	1059	(36.5)	1657	(46.3)	3807	(40.8)	5268	(44.8)
**Area Level Deprivation ^2,3^**										
Missing	1	(0.0)	1	(0.0)						
1 (least deprived)	2201	(8.0)	117	(4.0)	392	(10.9)	323	(3.5)	1369	(11.7)
2	2610	(9.5)	340	(11.7)	283	(7.9)	1135	(12.1)	852	(7.3)
3	4478	(16.2)	309	(10.6)	709	(19.8)	1097	(11.7)	2363	(20.1)
4	7180	(26.0)	670	(23.1)	976	(27.3)	2207	(23.6)	3327	(28.3)
5 (most deprived)	11,106	(40.3)	1468	(50.5)	1221	(34.1)	4580	(49.0)	3837	(32.7)
**District Health Board ^2,3^**										
Northland	6359	(23.1)	1539	(53.0)	-	-	4820	(51.6)	-	-
Waikato	15,329	(55.6)	-	-	3581	(100.0)	-	-	11,748	(100.0)
Hawke’s Bay	5888	(21.4)	1366	(47.0)	-	-	4522	(48.4)	-	-
**Model of maternity care at registration ^2,3^**										
DHB	322	(1.2)	120	(4.1)	0	(0.0)	201	(2.2)	1	(0.0)
General Practice	13	(0.0)	1	(0.0)	0	(0.0)	4	(0.0)	8	(0.1)
Midwife	26,367	(95.6)	2710	(93.3)	3486	(97.3)	8782	(94.0)	11,389	(96.9)
No LMC	817	(3.0)	71	(2.4)	88	(2.5)	341	(3.7)	317	(2.7)
Obstetrician	57	(0.2)	3	(0.1)	7	(0.2)	14	(0.1)	33	(0.3)
**Number of antenatal care visits ^2,3^**										
Missing	1385	(5.0)	218	(7.5)	110	(3.1)	618	(6.6)	439	(3.7)
0–5	4813	(17.5)	351	(12.1)	441	(12.3)	1208	(12.9)	2813	(23.9)
6–10	11,422	(41.4)	1191	(41.0)	1734	(48.4)	3557	(38.1)	4940	(42.0)
11–15	8896	(32.3)	1002	(34.5)	1188	(33.2)	3460	(37.0)	3246	(27.6)
16–20	941	(3.4)	126	(4.3)	99	(2.8)	434	(4.6)	282	(2.4)
21+	119	(0.4)	17	(0.6)	9	(0.3)	65	(0.7)	28	(0.2)
**Parity of eligible pregnancy ^1,3^**										
Missing	863	(3.1)	95	(3.3)	88	(2.5)	361	(3.9)	319	(2.7)
0	11,556	(41.9)	964	(33.2)	1277	(35.7)	3952	(42.3)	5363	(45.7)
1	7597	(27.5)	914	(31.5)	1158	(32.3)	2351	(25.2)	3174	(27.0)
2	4122	(14.9)	478	(16.5)	565	(15.8)	1420	(15.2)	1659	(14.1)
3	1844	(6.7)	219	(7.5)	274	(7.7)	663	(7.1)	688	(5.9)
4+	1594	(5.8)	235	(8.1)	219	(6.1)	595	(6.4)	545	(4.6)
**Gestational age at delivery (weeks) ^2,3^**										
28–38	7974	(28.9)	842	(29.0)	921	(25.7)	2736	(29.3)	3475	(29.6)
39–41	18,702	(67.8)	1998	(68.8)	2503	(69.9)	6410	(68.6)	7791	(66.3)
42–43	900	(3.3)	65	(2.2)	157	(4.4)	196	(2.1)	482	(4.1)

LMP = last menstrual period; LMC = lead maternity carer; DHB = District Health Board. Chi-square/fishers test for differences between control and intervention regions: in the pre-intervention period (^1^
*p* < 0.01, ^2^
*p* < 0.0001) and in the post-intervention period (^3^
*p* < 0.0001).

**Table 2 vaccines-10-00150-t002:** Pertussis vaccine uptake in the control and intervention groups pre- and post-intervention.

	Total		Pre-Intervention				Post-Intervention			
			Unvaccinated		Vaccinated		Unvaccinated		Vaccinated	
	*n*	(%)	*n*	(%)	*n*	(%)	*n*	(%)	*n*	(%)
Control areas ^1^	12,247	(100)	2138	(73.6)	767	(26.4)	5797	(62.1)	3545	(38.0)
Intervention area ^2^	15,329	(100)	2832	(79.1)	749	(20.9)	7636	(65.0)	4112	(35.0)

^1^ Control area refers to Northland DHB and Hawke’s Bay DHB where maternal pertussis vaccination was not funded in pharmacy. ^2^ Intervention area refers to Waikato DHB where the intervention was maternal pertussis vaccination funding extended to pharmacy.

**Table 3 vaccines-10-00150-t003:** The odds of receiving the pertussis vaccine during pregnancy in each study region in the post-intervention period compared with the pre-intervention period, overall and by ethnicity.

	Pre-Intervention		Post-Intervention		Adjusted Model ^1,2^	
	*n*	Proportion Vaccinated (95% CI)	*n*	Proportion Vaccinated (95% CI)	OR	(95% CI)
**Region ^2^**						
Control areas	2904	26.4 (24.5, 28.3)	9342	38.0 (36.7, 39.3)	1.67	(1.52, 1.84)
Intervention area	3581	20.9 (19.4, 22.4)	11,748	35.0 (33.9, 36.1)	2.07	(1.89, 2.27)
**Non-Māori ^3^**						
Control areas	1370	36.4 (33.2, 39.6)	4913	50.4 (48.4, 52.4)	1.79	(1.58, 2.03)
Intervention area	2292	27.4 (25.3, 29.5)	7698	42.7 (41.2, 44.2)	1.98	(1.79, 2.20)
**Māori ^3^**						
Control areas	1534	17.5 (15.4, 19.6)	4429	24.2 (22.8, 25.6)	1.50	(1.29, 1.74)
Intervention area	1289	9.4 (7.7, 11.1)	4050	20.4 (19.0, 21.8)	2.42	(1.99, 2.97)

CI Confidence interval. ^1^ Adjusted for maternal age at last menstrual period, maternal ethnicity, and maternal deprivation. ^2^ Significant regions (control/intervention) by time (pre/post) interaction (*p* = 0.0014). ^3^ Significant regions (control/intervention) by time (pre/post) by ethnicity (non-Māori/Māori) interaction (*p* = 0.0281).

## Data Availability

Data used in this study is publicly available from the New Zealand Ministry of Health.

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
