# Peer review of "Increasing Uptake of Maternal Pertussis Vaccinations through Funded Administration in Community Pharmacies"

_vaccines, 2022, doi:10.3390/vaccines10020150_

Round 1

Reviewer 1 Report

The manuscript from Howe et al., is about the impact of extending funding for maternal pertussis vaccination in community pharmacies in New Zealand. The manuscript is well written, description of results is short but accurate. I only have minor comments. 

-Please expand the logistical reasons mentioned in line 51.

-The study setting section could be reorganized and shortened. Also,  the paragraph line 96 to 100 could be moved up after line 62.

-Add the timeline of the study in the paragraph 59-61

-Please add "areas" to Control and intervention in all tables.

Author Response

The manuscript from Howe et al., is about the impact of extending funding for maternal pertussis vaccination in community pharmacies in New Zealand. The manuscript is well written, description of results is short but accurate. I only have minor comments.

Please expand the logistical reasons mentioned in line 51.

  • “(e.g cold chain)” has been added as requested

The study setting section could be reorganized and shortened. Also, the paragraph line 96 to 100 could be moved up after line 62.

  • This has been amended as requested

Add the timeline of the study in the paragraph 59-61

  • “over three years” has been added to the text here

Please add "areas" to Control and intervention in all tables

  • This is been done as requested

Reviewer 2 Report

In abstract line 18, it seems it should be "maternal pertussis vaccination" instead of "maternal pertussis".

In method, lines 157 to 159, "Area level socioeconomic deprivation (NZ Dep 2013 Index of Deprivation) was categorised into quintiles (1-5) with ‘1’ being the least deprived and ‘5’ the most deprived quintile and into binary (low = deciles 1-5 and high =deciles 6-10). " sentence needs improvement, it is not clear how the categorization has been performed, quintiles to deciles.

Please provide CI95% for each vaccine uptake measure in pre and post intervention in both groups in text and also in table  3.

I could not find the results of "autoregressive correlation structure accounting for repeated pregnancies", "generalised estimating equations with a binomial distribution and logit link" and "three-way interaction term" analysis as described in method. The results should be explicitly described and discussed in the discussion.

ORs for vaccine uptake change between groups can be calculated and can provide much more information for comparison, so it is strongly recommended to be reported and discussed.

A self-citation, reference 21, has not published yet but referenced.

Author Response

In abstract line 18, it seems it should be "maternal pertussis vaccination" instead of "maternal pertussis".

  • This has been amended as suggested

In method, lines 157 to 159, "Area level socioeconomic deprivation (NZ Dep 2013 Index of Deprivation) was categorised into quintiles (1-5) with ‘1’ being the least deprived and ‘5’ the most deprived quintile and into binary (low = deciles 1-5 and high =deciles 6-10). " sentence needs improvement, it is not clear how the categorization has been performed, quintiles to deciles.

  • The text was amended with the following: “Socio-economic deprivation was measured using an area level measure (NZ Dep 2013 Index of Deprivation), which was categorised from deciles (‘1’ = least deprived and ‘10’ = most deprived) into quintile (1-5) and binary (low = deciles 1-5 and high =deciles 6-10) variables.”

Please provide CI95% for each vaccine uptake measure in pre and post intervention in both groups in text and also in table  3.

  • This is been done as requested

I could not find the results of "autoregressive correlation structure accounting for repeated pregnancies", "generalised estimating equations with a binomial distribution and logit link" and "three-way interaction term" analysis as described in method. The results should be explicitly described and discussed in the discussion.

  • The following as been added as requested, “The secondary analysis including all of a woman’s eligible pregnancies did not changed the results (data not shown).”
  • Table 3 presents the results of the three-way interaction and was discussed in the following text, “Pertussis vaccine uptake for Māori women was lower than for non-Māori women, in both the pre- and post-intervention study periods (Table 3). The increase in pertussis vaccination uptake for Māori women was greater in the intervention region (OR=2.42, 95% CI 1.99, 2.97) than the control regions (OR=1.50, 95% CI 1.29, 1.74) (p=0.0281) (Table 3). There was no significant difference in pertussis vaccine uptake for non-Māori women between the control and intervention regions over the study period (Table 3). There was no significant difference in pertussis vaccine uptake by area-level socioeconomic deprivation, i.e., women living in more socioeconomically deprived areas had a similar proportional increase in uptake to women living in less socioeconomically deprived areas.”

ORs for vaccine uptake change between groups can be calculated and can provide much more information for comparison, so it is strongly recommended to be reported and discussed.

  • We respectfully decline to make this change. We have presented the ORs for change between the post and pre time periods by study region which allows the reader to determine the difference, and we have included the interaction term in our model that tests whether the difference in odds of vaccine post vs pre varies by region (intervention/control). The statistical significance of this interaction term (p=0.0014), which indicates that the change between the post and pre time periods differed significantly by study region, is also presented in Table 3.

A self-citation, reference 21, has not published yet but referenced.

  • This reference refers to a paper submitted to the same journal at the same time on the same study but from a qualitative point of review. This paper has also been reviewed and the reference will be updated when accepted.

Round 2

Reviewer 2 Report

Some changes performed according to the comments.

Remained major concern is related to one citation in this study which has not been published or accepted yet.

Decision: Reconsider after the status of reference no 21 will be confirmed.

Author Response

Ref #21 has been updated as requested